# The Effect of Waning on Antibody Levels and Memory B Cell Recall following SARS-CoV-2 Infection or Vaccination

**DOI:** 10.3390/vaccines10050696

**Published:** 2022-04-29

**Authors:** David Forgacs, Vanessa Silva-Moraes, Giuseppe A. Sautto, Hannah B. Hanley, Jasper L. Gattiker, Alexandria M. Jefferson, Ravindra Kolhe, Ted M. Ross

**Affiliations:** 1Center for Vaccines and Immunology, University of Georgia, Athens, GA 30605, USA; forgacs1@uga.edu (D.F.); vmoraes@uga.edu (V.S.-M.); gasautto@uga.edu (G.A.S.); hhanley2@uga.edu (H.B.H.); egatt@uga.edu (J.L.G.); alexandria.jefferson@uga.edu (A.M.J.); 2Department of Pathology, Medical College of Georgia, Augusta University, Augusta, GA 30912, USA; rkolhe@augusta.edu; 3Department of Infectious Diseases, University of Georgia, Athens, GA 30605, USA

**Keywords:** SARS-CoV-2, COVID-19, antibody decay, waning, vaccination, infection, pre-immunity, memory B cell recall, antibody secreting cells

## Abstract

In order to longitudinally track SARS-CoV-2 antibody levels after vaccination or infection, we assessed anti-RBD antibody levels in over 1000 people and found no significant decrease in antibody levels during the first 14 months after infection in unvaccinated participants, however, a significant waning of antibody levels was observed following vaccination. Participants who were pre-immune to SARS-CoV-2 prior to vaccination seroconverted to higher antibody levels, which were maintained at higher levels than in previously infected, unvaccinated participants. Older participants exhibited lower level of antibodies after vaccination, but a higher level after infection than younger people. The rate of antibody waning was not affected by pre-immunity or age. Participants who received a third dose of an mRNA vaccine not only increased their antibody levels ~14-fold, but also had ~3 times more antibodies compared to when they received their primary vaccine series. PBMC-derived memory B cells from 13 participants who lost all circulating antibodies were differentiated into antibody secreting cells (ASCs). There was a significant recall of memory B cell ASCs in the absence of serum antibodies in 5–8 of the 10 vaccinated participants, but not in any of the 3 infected participants, suggesting a strong connection between antibody levels and the effectiveness of memory B cell recall.

## 1. Introduction

In late 2019, Severe Acute Respiratory Syndrome Coronavirus 2 (SARS-CoV-2), the causative agent of COVID-19, emerged in Wuhan, China, and quickly spread across the world resulting in the ongoing COVID-19 pandemic [1]. According to the World Health Organization (WHO), as of April 2022, there have been over 500 million reported SARS-CoV-2 cases worldwide, and more than 11 billion COVID-19 vaccine doses have been administered. As the virus continues to evolve and adapt to humans, several viral variants of concern have emerged, and most likely, new variants will continue to emerge in the future that could result in improved viral fitness, replication, and transmission rates [2].

Antibody-mediated immunity is essential to mount a systemic immune response against SARS-CoV-2. The receptor binding domain (RBD) located on the tip of the spike glycoprotein is one of the major targets of neutralizing antibodies [3,4]. The virus uses the spike protein to bind to the ACE2 receptors on the surface of host cells in order to gain cell entry [3,4]. Therefore, several COVID-19 vaccines were developed based on the spike protein as an immunogen [5,6]. The Pfizer–BioNTech (New York, NY, USA) BNT162b2 and the Moderna mRNA-1273 vaccines both contain mRNA coding for the full-length spike protein [5,6]. These vaccines are administered intramuscularly as two doses 21 or 28 days apart, respectively [5,6]. A third, single-dose viral vector vaccine by Janssen also received emergency use authorization in the USA [7,8]. An individual is considered fully vaccinated 14 days after they completed the primary vaccine series [5,9].

Antispike antibodies increase significantly following vaccination [10,11]. However, there is a significant drop in antibody levels within the first few months following vaccination [12,13,14]. In order to keep the level of circulating antibodies high, a third (booster) dose of the Pfizer–BioNTech and Moderna vaccines was approved in the USA during the second half of 2021 [15]. While all Pfizer–BioNTech vaccine doses—including the third (booster) dose—contain 30 µg mRNA, the primary Moderna vaccine series contains 100 µg each, and the booster dose contains only 50 µg [5,9,15,16].

Besides antibody-driven immunity, cellular immunity is also essential for long-term protection against infection [17,18]. The T cell response is crucial in initiating immune-mediated death of infected cells, coordinating the rest of the immune response by releasing cytokines, and activating memory B cells [19]. Memory B cells and long-lived plasma cells elicited by infection or vaccination can be recalled in the event of a repeat encounter, yielding a faster and more robust antibody response [17]. The relationship between the levels of circulating and memory recall response is an underexplored area of SARS-CoV-2 immunology research.

In 2020, SPARTA (SARS SeroPrevalence and Respiratory Tract Assessment), a study funded by the U.S. National Institutes of Health (NIH), was initiated to understand if immunity elicited following infection with SARS-CoV-2 or vaccination with COVID-19 vaccines provides protection against future infection and symptomatic disease [10]. The goals were (1) to investigate the level and duration of protection afforded by natural infection following SARS-CoV-2 infection, (2) to assess immunological risk factors for infection outcome and examine immune responses to infection across the disease spectrum, and (3) to study the immune effectiveness of COVID-19 vaccines in both pre-immune and immunologically naïve participants. Participants were located at several sites in three USA states and represented a diverse cohort that provided a real-time snapshot into the progression of immune responses during the pandemic. To date, ~3800 participants have provided serum, peripheral blood cells, and saliva at multiple timepoints.

One of the goals of this study was to examine the rate of antibody waning in a large cohort of people infected with SARS-CoV-2 and/or vaccinated with a commercial COVID-19 vaccine, as well as to discover differences in trends associated with antibody levels based on vaccination, infection, and pre-immunity. In addition, in order to ascertain whether the presence of serum antibodies is indicative of a robust recall response, peripheral blood mononuclear cells (PBMCs) were collected from 13 people whose memory B cells were differentiated in vitro into antibody secreting cells (ASCs). These 13 participants had an initial antibody response induced by vaccination or infection, but their antibody levels subsequently declined to undetectable levels. In the absence of serological protection, the de novo memory B cell recall response was analyzed in these participants.

## 2. Materials and Methods

### 2.1. SPARTA Participants

Eligible participants between 18 and 90 years old were enrolled starting in October 2020 with written informed consent in Athens and Augusta, GA; Memphis, TN; and Los Angeles, CA. The study procedures, informed consent, and data collection documents were reviewed and approved by the WIRB Copernicus Group Institutional Review Board (WCG IRB #202029060). Of the ~3800 SPARTA enrolled participants, 1064 were randomly selected from the Athens, GA, and Augusta, GA, cohorts to be included in this study (Appendix A); 68.7% of them identified as females, and 31.1% as males. The average age was 44.9 years (median age = 44 years old, SD = 17.2 years). Within the cohort, 86.2% identified as White, 7.2% as Black/African American, 4.1% as Asian, and 1.6% as multiple races; 7.3% of the participants were Hispanic or Latino. The range of BMI was between 17.5 and 95.5, with an average BMI of 28.5 (median BMI = 27.2, SD = 6.9).

### 2.2. Enzyme-Linked Immunosorbent Assay (ELISA)

ELISA assays were performed as previously described [10]. Briefly, Immulon^®^ 4HBX (Thermo Fisher Scientific, Waltham, MA, USA) or Costar EIA/RIA (Corning, Corning, NY, USA) plates were coated with 100 ng/well of recombinant SARS-CoV-2 RBD protein (based on USA/WA1/2020 strain), incubated with heat inactivated serum samples at a starting dilution of 1:50, and then further serially diluted 3-fold [20]. IgG antibodies were detected using horseradish peroxidase (HRP)-conjugated goat antihuman IgG detection antibody (Southern Biotech, Birmingham, AL, USA) at a 1:4000 dilution, and colorimetric development was accomplished using 100 μL of 0.1% 2,2′-azino-bis(3-ethylbenzothiazoline-6-sulphonic acid) (ABTS, Bioworld, Dublin, OH, USA) solution with 0.05% H_2_O_2_ for 18 min at 37 °C. The reaction was terminated with 50 μL of 1% (*w/v*) SDS (VWR International, Radnor, PA, USA). Colorimetric absorbance was measured at 414 nm using a PowerWaveXS plate reader (Biotek, Winooski, VT, USA). All samples and controls were run in duplicate, and the mean of the two blank-adjusted optical density (OD) values were used in downstream analyses. IgG equivalent concentrations were calculated based on a 7-point standard curve generated by a human IgG reference protein from plasma (Athens Research and Technology, Athens, GA, USA), and verified on each plate using human sera of known concentrations.

### 2.3. Viral Neutralization Assay

Viral neutralization (VN) assays were performed in a Biosafety Level 3 (BSL-3) laboratory, as previously described [10]. The USA-WA1/2020 SARS-CoV-2 strain (100 TCID50/50 µL; NCBI accession number: PRJNA717311) was co-incubated with serially diluted serum samples for 1 h at 37 °C and then added to a monolayer of Vero E6 cells. The plates were observed after 72 h for cytopathic effects (CPEs). The VN endpoint titer was determined as the reciprocal of the highest dilution that completely inhibited CPE formation. All neutralization titers were represented as the average of the three replicates.

### 2.4. In Vitro Differentiation

PBMC-derived memory B cells were differentiated into antibody secreting cells by incubating them with 500 ng/mL R848 (Invivogen, San Diego, CA, USA) and 5 ng/mL rIL-2 (R&D, Minneapolis, MN, USA) for 7 days at 37 °C in 5% CO_2_, as previously described [21]. R848, also known as Resiquimod, is a dual TLR7/8 agonist that activates B and T cells by way of the TLR7/8 MyD88-dependent signaling pathway [22], while IL-2 promotes the differentiation of T cells, which in turn activates the differentiation of B cells as well [23]. The conditioned cell culture supernatants were serially diluted to assess total and antigen-specific IgG antibody levels by ELISA. Total (nonantigen-specific) IgG levels were assessed to confirm that the in vitro differentiation was successful and B cells were actually recalled de novo. Antigen-specific IgG levels to SARS-CoV-2 RBD, and full-length spike protein (based on USA/WA1/2020 strain), as well as pandemic influenza strain A/H1N1/California/2009 (Cal/09) were also measured, along with spike-specific IgA and IgM levels. The experiment was performed twice using two separate PBMC vials.

### 2.5. B Cell FluoroSpot Assay

The number of total and spike-specific ASCs was determined using the human IgA/IgG/IgM three-color FluoroSpot kit from CTL (Cellular Technology Limited LLC, Cleveland, OH, USA) based on the instructions provided by the manufacturer. Briefly, high-binding PVDF filter plates were coated with antihuman Igκ/λ mixture (for total Ig plates) or 25 µg/mL of full-length spike protein (for antigen-specific plates). After in vitro differentiation, cells were serially diluted 3-fold starting at 2 × 10^4^ or 1 × 10^5^ cells, respectively, in triplicates. Plates were incubated for 16 h at 37 °C, in 5% CO_2_. Plates were then washed and incubated in the dark with a mixture of secondary IgA/IgG/IgM detection antibodies for an additional 16 h. Plates were then washed and incubated with SA-CTL-Red^TM^ and Anti-Hapten CTL-Yellow^TM^ tertiary antibody solution for 1 h at room temperature (RT). Next, the plates were decanted, washed, and the membrane was allowed to completely dry in the dark. Plates were scanned using an S6 Ultimate M2 ImmunoSpot reader and counted using the ImmunoSpot 7.0.28.5 Analyzer software (Cellular Technology Limited LLC, Cleveland, OH, USA).

### 2.6. Direct Ex Vivo B Cell Immune Cell Phenotyping

Cryopreserved cells were thawed by diluting them in 10 mL pre-warmed complete B cell media (RPMI + 5% FBS + 1% Pen/Strep) in the presence of DNAse (20 µg/mL) and spun at 500× *g* for 10 min. Supernatants were carefully removed, and the cells were counted and rested at 2 × 10^6^ cells/mL for 3 h. After resting, cells were washed again in complete B cell media and 1 × 10^6^ cells were resuspended in 50 µL of PBS containing LIVE/DEAD (Thermo Scientific, Waltham, MA, USA) and human FC block (1:50, Biolegend), and incubated for 20 min at RT. After incubation, cells were topped with 150 µL of PBS with 2% FBS (FACS buffer) and spun 500× *g* for 4 min. The supernatant was removed and the antibody mix containing the surface antibodies was added to the cells and incubated for 30 min at RT in the dark (Appendix A). Following surface staining, cells were washed twice with FACS buffer and resuspended in a final volume of 200 µL. Samples were acquired on Agilent NovoCyte Quanteon and analyzed using FlowJo v10.2.

### 2.7. Statistical Analysis

Comparison of the four groups (naïve unvaccinated, infected unvaccinated, naïve vaccinated, infected vaccinated) based on previous infection and vaccination as well as different timepoints were investigated by one-way ANOVA and paired *t*-tests using GraphPad Prism 9.3.1 (RRID: SCR_002798). In order to compare the slopes of the four groups, the antibody concentrations were log-transformed, and the slopes after a linear regression analysis were compared by ANOVA. Statistical significance was denoted as * *p* < 0.05, ** *p* < 0.01, *** *p* < 0.001, and **** *p* < 0.0001; *p* > 0.05 was considered not significant (ns). Participants infected or vaccinated during the course of the study had their relevant timepoints included in multiple categories. For naïve unvaccinated participants, their changes in antibody titer were tracked starting with their earliest available timepoint. For infected unvaccinated participants, tracking started two weeks (0.5 months) after positive COVID nucleic acid amplification test (NAAT) or symptom onset; while for vaccinated participants (regardless of pre-immunity), tracking started two weeks (0.5 months) after receiving the complete primary series of vaccines to allow time for seroconversion. Completing the primary series is defined as having received two doses of the Pfizer–BioNTech or the Moderna vaccines or one dose of Johnson & Johnson’s Janssen vaccine.

## 3. Results

In order to track and compare the changes in anti-RBD IgG antibody levels, the anti-RBD IgG antibody concentrations of 1064 participants were tracked, based on a total of 3932 individual timepoints (Figure 1, Appendix A). Participants were divided into four distinct categories: (1) The naïve, unvaccinated group (n = 418) included participants who were never infected with the SARS-CoV-2 virus or vaccinated. (2) The infected, unvaccinated group (n = 290) comprises participants who had a confirmed SARS-CoV-2 infection either by NAAT, rapid antigen test, or a combination of COVID-19-specific symptoms followed by a corresponding significant rise in anti-RBD antibodies. (3) The naïve, vaccinated group (n = 515) encompasses participants who were never infected with the SARS-CoV-2 virus but received their primary vaccine series. (4) The infected, vaccinated group (n = 298) comprises participants who were pre-immune to the SARS-CoV-2 virus at the time of their vaccination and received their primary vaccine series.

The naïve, unvaccinated participants had antibody levels (mean = 0.4 µg/mL) that were below the experimentally determined concentration threshold and did not significantly change over time (*p* = 0.19) (Figure 1) [10]. Participants who were infected but unvaccinated seroconverted with an average initial IgG antibody level of 3.9 µg/mL. There was no significant waning of antibody levels in these participants (*p* = 0.0876) over the first 14 months following infection (mean concentration across all timepoints = 3.5 µg/mL). There was a much greater initial mean antibody level (44.8 µg/mL) in naïve vaccinated participants; however, this concentration significantly decreased (**** *p* < 0.0001) each month for the first 6 months. The infected vaccinated group initially seroconverted even higher (mean = 85 µg/mL); however, the antibody level experienced significant waning (**** *p* < 0.0001) each month for the first 5 months (Figure 1).

The anti-RBD IgG antibody concentrations of the naïve unvaccinated group were significantly lower than the antibody concentrations observed in any of the other three groups at all timepoints (**** *p* < 0.0001) (Figure 1). Participants who were infected and then vaccinated had significantly higher antibody levels than participants in the other three groups (**** *p* < 0.0001 compared to infected unvaccinated and naïve unvaccinated groups, ** *p* < 0.042 compared to naïve vaccinated group). Naïve participants who were vaccinated had significantly higher anti-RBD antibody concentrations compared to unvaccinated participants who were previously infected for the first four months (**** *p* < 0.0001 for the first three months, ** *p* = 0.0016 during the fourth month) and no longer showed a significant difference beyond that timepoint (Figure 1). The slopes of each of the four groups were significantly different from every other group (**** *p* < 0.0001) with the exception of the two unvaccinated groups (*p* = 0.9569) and the two vaccinated groups (*p* = 0.7914).

The slope for each group was not significantly different between young (18–39 years old) and older (50 and older) participants (Figure 2A). However, the two age groups differed considerably in their initial magnitude of seroconversion (0.5–1 month timepoint). Young naïve participants seroconverted to significantly higher levels after vaccination (55.5 vs. 38.7 µg/mL, ** *p* = 0.0038), while infected and unvaccinated older participants demonstrated higher magnitude of seroconversion on average (10.86 vs. 3.13 µg/mL, * *p* = 0.0334 (Figure 2A). There was no significant difference between young and old participants in the naïve, unvaccinated (*p* = 0.1074) and infected, vaccinated groups (*p* = 0.789) (Figure 2A).

There was no difference in the slope of waning antibodies or the magnitude of initial seroconversion between Pfizer- and Moderna-vaccinated participants in either the naïve, vaccinated or the infected, vaccinated groups (Figure 2B).

The naïve, vaccinated group was the only cohort that demonstrated a significant difference in the slope of antibody waning (* *p* = 0.0318) and the magnitude of initial seroconversion (* *p* = 0.0438) between males and females (Figure 2C). Females in that group showed higher initial antibody levels after vaccination (48 vs. 35.62 µg/mL), but also a faster rate of waning (Figure 2C).

The naïve, vaccinated group was the only cohort that demonstrated a significant difference in the slope of waning antibodies (** *p* = 0.0039) and the magnitude of initial seroconversion (* *p* = 0.0119) between participants with a BMI in the healthy range (18.5–24.9) and those who were obese (30+) (Figure 2D) [24]. Obese participants in that group showed higher initial antibody levels after vaccination (52.5 vs. 35.86 µg/mL), but also a faster rate of waning (Figure 2D).

Out of the 1064 participants randomly selected for the longitudinal serum analysis, the antibody responses of participants who received two base mRNA vaccinations and an mRNA booster vaccine dose were assessed in 306 participants (Figure 3). Most of the participants received a homologous booster (227 Pfizer–BioNTech and 55 Moderna), but 24 received a heterologous booster vaccine dose (13 with Pfizer base + Moderna booster, 11 Moderna base + Pfizer booster). On average, participants experienced a 14-fold increase in their antibody levels compared to their last pre-boost timepoint (**** *p* < 0.0001), reaching a mean antibody concentration of 119.2 µg/mL (Figure 3). Moreover, their antibody levels were also on average three times higher than initially 2–4 weeks after receiving their second base vaccine dose (**** *p* < 0.0001), which was on average 39.8 µg/mL and remained significantly higher for the first 3 months (** *p* = 0.0012). After the reception of the booster dose, significant waning was observed (**** *p* < 0.0001 between the first and second month, * *p* < 0.05 between the second and third month), but the antibody levels were still significantly higher 5 months after the booster compared to the level immediately preceding the booster (** *p* = 0.0067).

Out of the entire cohort, a total of 25 participants were identified who at some point during the study tested positive for RBD-binding IgG antibodies due to vaccination or infection, but have waned below the threshold into the negative range on a later timepoint (Figure 4A, Appendix A). In order to confirm that these participants have in fact lost seroprotective status, viral neutralization tests using infectious SARS-CoV-2 USA-WA1/2020 strain were performed. Out of the 25 candidates, 12 either had nondetectable neutralization at the pre-waned timepoint (false positive by ELISA) or still had some level of neutralization potential on the post-waned timepoint (false negative by ELISA) (Figure 4B). The other 13 participants demonstrated some level of neutralization at their pre-waned timepoints but no neutralization at their post-waned timepoint, confirming the ELISA results that the participants have in fact lost their pre-existing seroprotection (Figure 4B). PBMCs collected from the post-waned timepoint from these 13 participants (10 post-vaccination and 3 post-infection) underwent in vitro differentiation, wherein they were stimulated by recombinant IL-2 and R848 to induce a memory B cell recall response. In addition, two normal converters (CVI-004 and P-073)—participants who seroconverted after vaccination as expected—and four nonconverters—participants who never experienced a significant increase in antibody levels after vaccination or infection—were also included as controls. The normal converters had a significant level of total IgG, Cal/09 IgG, RBD IgG, and spike IgG antibodies, while the nonconverters only had a significant level of total IgG antibodies, signifying that the experiment was successful at eliciting the antibody secretion (Table 1). Of the 10 participants who lost seroprotected status post-vaccination, 5–8 had a significant recall response against RBD and spike, while the other 2–5 did not (Table 1). The range is due to differences between the two times the experiment was performed using different PBMC vials (denoted as +/− in Table 1). Two showed no significant Cal/09 antibody levels. None of the three participants who lost seroprotected status post-infection demonstrated a significant recall response against RBD or spike, while two out of three showed a significant Cal/09 response (Table 1).

The number of ASCs was enumerated using a FluoroSpot assay. The number of spike-specific ASCs out of 10^6^ PBMCs was between 0 and 210 (Figure 5A), which represents 0–0.7% of total ASCs (Figure 5B). The majority of secreted immunoglobulins for each sample was IgG, and the spike-specific and total IgG/IgA/IgM ASCs of selected participants from each of the four groups are represented in Figure 5C. The only differences between the post in vitro differentiation ELISA results and the FluoroSpot results were three participants who had detectable total IgG antibodies by ELISA but no detectable ASCs by FluoroSpot. The antispike antibodies correlated with the presence/absence of spike-specific ASCs in all cases, but CVI-237, CVI-256, and CVI-519 showed a dissimilarity with the results of an earlier experiment using a different PBMC vial.

Because of limited availability of PBMCs, B cell immunophenotyping could only be performed on seven participants who waned post-vaccination (Appendix A). Six of the seven were from participants who lost seroprotection post-vaccination and showed a significant SARS-CoV-2-specific memory recall based on ELISA, while the seventh was a nonconverter (CVI-732) who lacked all antigen-specific B cells after in vitro stimulation. While 20.4% of CD19^+^ CD20^+^ cells were memory B cells in the first six participants ranging from 7% to 21% of IgG^+^ cells (median = 11.7%), CVI-732 demonstrated no detectable CD19^+^ B cells by flow cytometry (Appendix A).

## 4. Discussion

A few months after the WHO declared the global COVID-19 pandemic in the spring of 2020 [25], participants were enrolled in the SPARTA program and tracked for immune responses elicited by SARS-CoV-2 infection. Longitudinal samples were collected from over 1000 participants before and after infection and subsequently following vaccination. The data in this report analyzes the immune responses prior to the Omicron variant wave that began in December 2021.

There was a stark difference in the waning of antibody levels in vaccinated versus infected participants. While infection-induced antibody levels were ~10-fold lower than vaccinated participants, there was no significant waning over the next 14 months after infection. While vaccination-induced antibody levels measured 2–4 weeks post-vaccination were significantly higher than infection-induced antibodies, they waned quickly, eventually asymptoting to levels similar to participants who were infected but not vaccinated. Thus, the difference between the antibody levels in infected participants regardless of vaccination status is only apparent in the primary response and diminishes in the following months. Participants who were both infected and vaccinated had significantly higher antibody levels than infected unvaccinated participants for over 10 months. In contrast, participants who were never infected with SARS-CoV-2 before vaccination had significantly higher antibody levels compared to infected but unvaccinated participants for only the first 4 months following vaccination. In line with previous observations, the most robust antibody levels were reached and maintained by participants who were infected prior to vaccination [10,26]. However, there was no significant difference between the rate of waning between the naïve vaccinated and the infected vaccinated groups, suggesting that pre-immunity does not significantly affect the rate of the waning of antibodies after vaccination. Thus, the differences between the two vaccinated groups were entirely due to the higher level of initial seroconversion amongst pre-immune participants. As the half-life of human IgG antibodies is 2–4 weeks, it is evident that the level of waning is proportionally related to the antibody level, and as their antibody levels dwindle, they asymptote instead of continuing to decline in a linear fashion [27,28].

Magnitude of seroconversion after vaccination in naïve participants was 1.4-fold lower in older compared to younger people, as previously shown [29], whereas older participants seroconverted to 3.5-fold higher antibody levels after infection (Figure 2A). There was no significant age-related difference between seroconversion levels in participants who were infected and vaccinated, suggesting that after vaccination, a young immune system with higher plasticity compensates for the poorer seroconversion after infection to reach comparable levels to older people. The lack of immunosenescence in infected participants could possibly be due to their relative latency in clearing the virus, causing a more prolonged acute immune reaction after infection [30].

Antibody waning following mRNA vaccination has been previously reported [31,32,33,34], and booster vaccine doses were approved in order to help to keep circulating antibody levels high. However, antibody waning may not be relevant, since a high level of circulating antibodies is not necessary for an effective B memory recall response [13,35,36]. In order to explore this dichotomy, this study assessed the effect of administering a third mRNA vaccine dose on antibody levels, as well as the memory B cell recall response in participants with undetectable levels of anti-SARS-CoV-2 antibody levels following the initial presence of antibodies after vaccination.

Participants who received a booster vaccination had a 14-fold increase in anti-RBD antibody levels, and their post-booster antibody levels were three times higher than it was after receiving their second dose. This suggests that not only do boosters effectively increase the quantity of circulating serum antibodies, but they also generally provide more antibodies capable of binding to the RBD region of the spike protein. While there is still significant waning following the booster dose, antibody levels stay higher for a longer period of time.

Memory B cell recall occurs in the vast majority of SARS-CoV-2 vaccinated and convalescent individuals [37,38,39,40]. While the waning of antibody levels is natural due to the short half-life of antibodies [27,28], concerns were raised about what happens to a very specific subset of participants who had a blunted serological response demonstrated by significantly lower initial antibody concentrations after vaccination (* *p* = 0.026; mean for cohort = 44.8 µg/mL, mean for subset = 11.77 µg/mL). This subset of participants subsequently waned below the level of seroprotection [13]. In order to explore the effectiveness of B cell memory recall in the absence of detectable circulating antibodies, the SARS-CoV-2 spike-specific recall response in differentiated PBMCs from participants who lost their seroprotected antibody levels after vaccination or infection was assessed. Though the anti-RBD ELISA had a sensitivity and specificity of 96% [10], most of these participants were close to the threshold and had a higher likelihood of being categorized as false positive or false negative. In order to verify that they did indeed possess protective antibodies, which later declined to undetectable levels, VN assays were performed in order to confirm seroprotected status. While analysis of binding antibody levels is useful and highly correlated with neutralization titers [10,41,42,43], VN assays using naturally occurring infectious SARS-CoV-2 are useful to assess not only the ability of antibodies to bind a small portion of the virus but also to quantify all antibodies capable of preventing cytopathic effects elicited by the entire live virus, creating a more realistic in vitro model.

Out of the 25 participants whose anti-RBD binding antibodies declined below the threshold of protection based on ELISA, 13 also lost their previous ability to neutralize the virus (Appendix A). The other 12 participants had either a false negative or a false positive ELISA result. Even though the sensitivity and specificity of the ELISA assay are 96%, participants whose antibody concentrations approach the cutoff have an increased chance of being falsely categorized due to the natural fluctuations of antibody levels, as well as minor inaccuracies in conducting the assay. In addition, there is a chance that the majority of antibodies produced by ELISA-negative VN-positive participants were directed at non-RBD antigenic targets, including other regions of the spike or nucleocapsid protein, which would explain why the anti-RBD ELISA was negative but there was a positive VN titer. In addition to the 13 participants who seroreverted, two normal vaccine responders and four who never seroconverted after vaccination were included as controls. After in vitro PBMC stimulation with IL-2 and R848, all normal responders were able to recall SARS-CoV-2-specific antibodies, while the nonresponders did not have a significant SARS-CoV-2 antigen-specific recall, but still produced nonspecific IgG antibodies de novo (Table 1). Three of the four nonresponders had autoimmune conditions or were taking immunosuppressant medication (Appendix A). Of the 13 participants who lost previously existing seroprotected status, 10 initially seroconverted due to vaccination, while the other three initially seroconverted due to a SARS-CoV-2 infection. All three had asymptomatic infections. None of the three previously infected and only 5–8 out of 10 vaccinated participants showed a significant SARS-CoV-2-specific recall response based on ELISA. B cell immunophenotyping confirmed that six of the participants who had a robust memory recall had a significant number of IgG^+^ memory B cells (Appendix A). On the other hand, CVI-732, a nonresponder taking the corticosteroid medication Prednisone [44], along with the anti-CD20 monoclonal therapy rituximab [45,46] for rheumatoid arthritis, had no antigen-specific antibody recall post-stimulation and no detectable B cells by flow cytometry (Table 1, Appendix A). Future studies will focus on the effect immunosuppressants and autoimmune conditions have on serological and cellular responses.

Enumerating the number of ASCs for these participants has led to similar conclusions as the in vitro differentiation and the ex vivo B cell immune cell phenotyping (Table 1). While there were a few instances where the ELISAs and one instance where the FluoroSpot assay was more sensitive, this can be due to the lower number of cells interrogated for each participant (4.5 × 10^5^ for spike-specific plates, 9 × 10^4^ for total Ig plates) in the FluoroSpot assay or that dying cells could have released antibodies that are detected by ELISA (Table 1). While the normal responders yielded positive results by both methods, no IgG ASCs could be detected by FluoroSpot amongst the cells of nonresponders, while total IgG antibodies were detectable (albeit at lower levels) by ELISA. CVI-732, one such nonresponder was also the participant with no detectable B cells by flow cytometry, further suggesting that the antibodies by ELISA may be detecting antibodies released from dead cells. In the group of participants who lost seroprotection post-vaccination, three participants had contrasting results between two different PBMC vials used for ELISA, causing the percent of participants who mounted a B cell recall response in the absence of circulating antibodies to range from 50 to 80% (5–8 out of 10). The PBMC vial that was used for both FluoroSpot and ELISA yielded congruent results in all cases between spike-specific ASCs and antibodies. There was no difference in the group of participants who lost seroprotection post-infection between the two methods.

In previous studies, there have been reports of blunted recall response after infection with coronaviruses. One such study from 2011 detected no memory B cells in any of the 23 people who recovered from SARS-CoV infection, while a memory T cells response was present in more than half [47]. Another study found that SARS-CoV-2 may blunt the germinal center response due to a block in Bcl-6^+^ T_FH_ cell differentiation, an increase in T-bet^+^ T_H1_ cells, and aberrantly high TNF-α levels, causing a diminished recall response [48]. Our finding that antibody recall to Cal/09 was also absent in all nonresponders and in some who lost seroprotection to SARS-CoV-2 suggests that while the overall IgG recall is substantial, the antigen-specific recall response in these participants is blunted not only against SARS-CoV-2 but also the Cal/09 pandemic influenza strain, perhaps due to overall lower level of total IgG antibodies (Table 1).

In this study, the IgG subclass composition was not determined. Previous studies identified that both SARS-CoV-2 infection and vaccination induce primarily an IgG1/IgG3-driven antibody response [49,50]. IgG1 antibodies specialize in binding and neutralizing antigens, while IgG3 antibodies are best at activating the complement cascade and ADCC [50]. Cytokines that drive class switch recombination are an important factor in the complement cascade, and helper T cells are necessary for the recall of memory B cells. Skewed proportions leading to the over- or under-representation of specific IgG subclasses that are directed at certain SARS-CoV-2 antigen targets have been associated with a more severe disease outcome [50] and some suggested that a disproportional response of IgG3 antibodies are associated with more severe or prolonged COVID-19 disease [51,52,53]. Therefore, it is possible that some nonconverters, as well as seroreverters in this study, may have also demonstrated signs of subclass/target preference different from the norm.

One of the limitations of this study is that we relied on accurate reporting of demographic information, comorbidities, previous SARS-CoV-2 testing and symptoms, and vaccine information by the participants. The causative strain for each infected participant is unknown and could come from the original SARS-CoV-2 strain or any pre-Omicron variants including Delta. Our pilot studies did not show significant differences between antibody binding to original strain vs. variant RBD/spike proteins, but drifted variants such as Omicron or future variants may cause a higher rate of antibody decay due to reduced avidity, leading to higher dissociation rates. While the sample size for each of the four infected/vaccinated groups compared is robust, not all timepoints were available for all participants. Only participants who were convalescent prior to vaccination were reported in the infected vaccinated group, participants who had breakthrough infections after vaccination will be included in a future study. The sample size for the in vitro differentiation experiment was low due to the uncommon nature of losing seroprotective status amongst our participants. All participants who lost their confirmed seroprotected status were included in the analysis; future studies are necessary to strengthen our findings with additional participants. The cohort was skewed towards White (86.2%) and non-Hispanic or Latino (92.7%) participants, which limits the conclusions we can make based on other racial or ethnic groups.

## 5. Conclusions

Vaccinated groups seroconverted to higher antibody levels and experienced significant antibody waning regardless of pre-immunity. In contrast, infected unvaccinated participants had no significant waning during the first 14 months after infection. The rate of antibody waning was not significantly different between pre-immune and immunologically naïve participants. Antibody levels are greatly increased by the administration of a booster dose, to a level three-fold higher than their previous peak antibody level 2–4 weeks after the second dose. Memory B cell recall responses were absent in all infected and 20–50% of vaccinated participants who lost seroprotected status. Thus, the relationship between circulating anti-SARS-CoV-2 antibody levels and B cell memory recall may be more tightly linked than expected. Therefore, maintaining high circulating antibody levels by administering booster doses could be a highly effective way to counteract antibody waning and avoid excessive waning of antibody levels.

## Figures and Tables

**Figure 1 vaccines-10-00696-f001:**
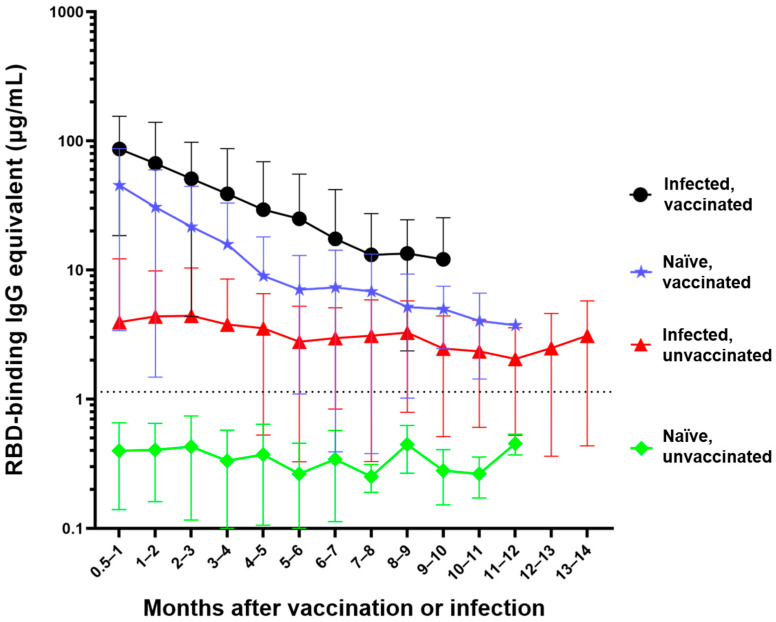
Differential waning of RBD-binding IgG antibody levels based on vaccination and infection status. Naïve unvaccinated (n = 418) and infected unvaccinated (n = 290) show no change in antibody levels over time (*p* > 0.05); naïve, vaccinated participants (n = 515) and infected, vaccinated participants (n = 298) both show significant waning over the time (**** *p* < 0.0001). The antibody level of the naïve unvaccinated group was always lower than the other groups (**** *p* < 0.0001); the infected vaccinated group was always higher than any other group (** *p* < 0.0042); naïve, vaccinated group is higher than infected, unvaccinated group for the first 4 months after vaccination (** *p* < 0.0016). The rate of decay was only significantly different between the vaccinated and infected groups (**** *p* < 0.0001) but not between the two vaccinated (*p* = 0.7914) and the two unvaccinated groups (*p* = 0.9569). Number of months start with the time of reception of the primary vaccine series for the vaccinated groups, time of infection for the infected unvaccinated group, and the first available timepoint for the naïve unvaccinated group.

**Figure 2 vaccines-10-00696-f002:**
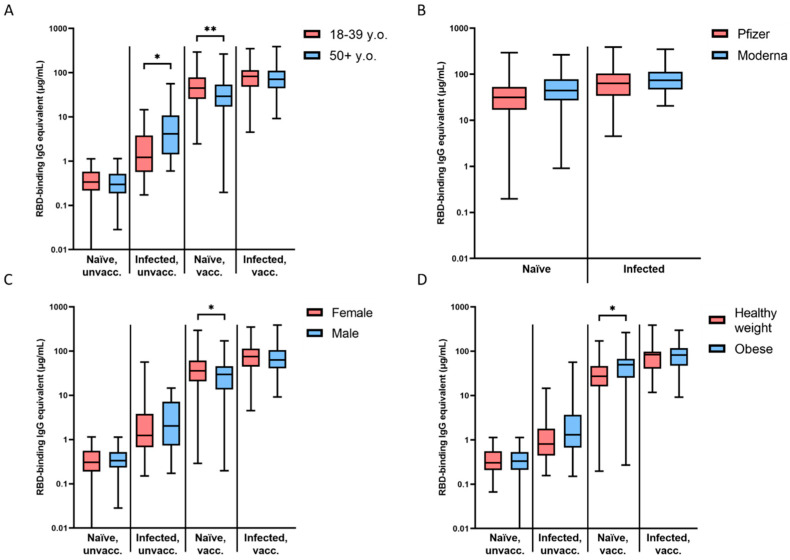
Magnitude of initial seroconversion based on demographic information. (**A**) Differences between young (18–39 y.o.) and older (50+ y.o.) participants were significant in infected, unvaccinated (* *p* = 0.0334) and naïve, vaccinated (** *p* = 0.0038) groups. (**B**) Differences between Pfizer and Moderna vaccinees were not significant. (**C**) Differences between females and males were significant in the naïve, vaccinated (* *p* = 0.0438) group. (**D**) Differences between participants with a healthy weight (BMI = 18.5–24.9) and those who were obese (BMI ≥ 30) were significant in the naïve, vaccinated group (* *p* = 0.0119). Initial seroconversion is designated by timepoints 0.5–1 month after COVID-19 infection or the reception of the primary vaccine series (2 × mRNA or 1 × viral vector).

**Figure 3 vaccines-10-00696-f003:**
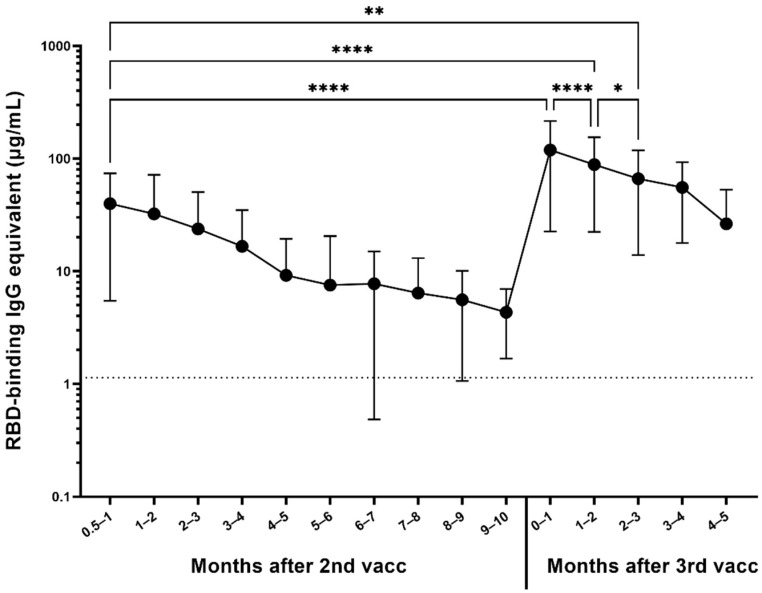
Response to 3rd dose of mRNA vaccines (n = 306). Participants are represented regardless of pre-immunity. Significant increase in anti-RBD IgG antibody level was observed between the last available pre-booster timepoint and the timepoint within the first month after the booster (**** *p* < 0.0001). The difference between the timepoint 2–4 weeks after the second dose and the timepoint within the first month after the booster was also significant (**** *p* < 0.0001). Month-to-month antibody decay is significant for the first 3 months after the booster (**** *p* < 0.0001 for the for the first two months, * *p* < 0.05 between the second and the third month), at which point the antibody level was still significantly higher than 2–4 weeks after the second dose (** *p* = 0.0012).

**Figure 4 vaccines-10-00696-f004:**
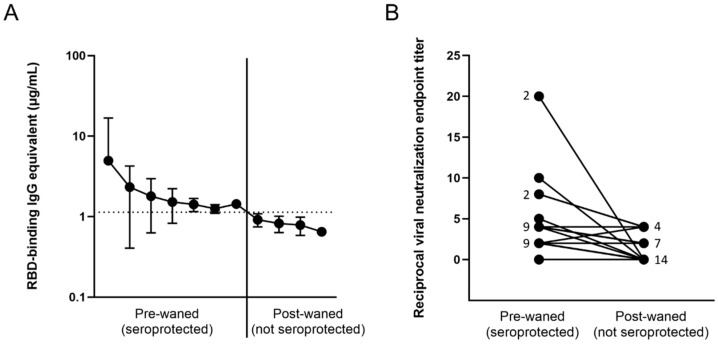
Loss of antibody-protected status based on (**A**) antibody binding, and (**B**) viral neutralization in waned participants. (**A**) All 25 participants demonstrated a loss of anti-RBD binding antibodies; (**B**) only 13 showed a loss of neutralization potential, signified by an initial non-0 neutralization endpoint titer and an endpoint titer of 0 at the post-waned timepoint. The numbers next to the dots represent the number of overlapping participants with the same neutralizing endpoint titer.

**Figure 5 vaccines-10-00696-f005:**
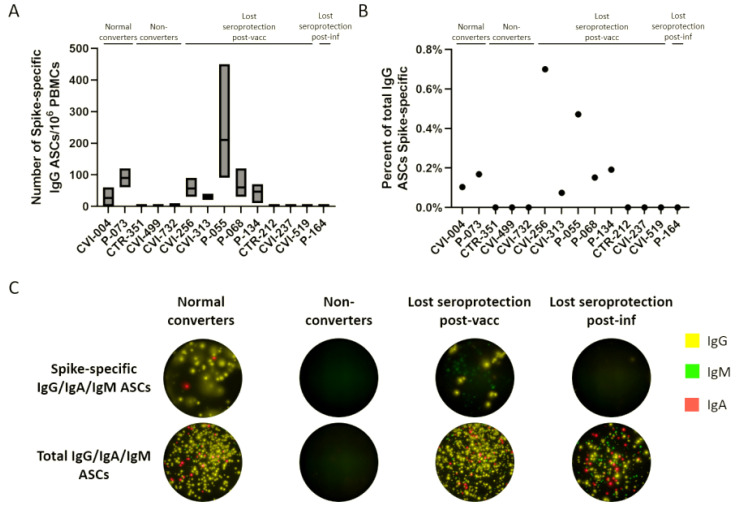
ASC enumeration in nonresponders and participants who lost seroprotection post-vaccination or infection. (**A**) Number of spike-specific IgG ASCs out of 10^6^ in vitro differentiated PBMCs. (**B**) Percent of total IgG ASCs that are spike-specific. (**C**) Representative wells for normal converters, nonconverters, and participants who lost seroprotection post-vaccination or infection, for spike-specific and total ASCs.

**Table 1 vaccines-10-00696-t001:** Memory B cell recall response in participants who lost their seropositive status after vaccination or infection.

	Participant	Total IgG	Total ASCs	Cal/09 IgG	RBD IgG	Spike IgG	Spike ASCs
**Normal responders**	CVI-004	+	+	+	+	+	+
	P-073	+	+	+	+	+	+
**Non-responders**	CTR-351	+	−	−	−	−	−
	CVI-499	+	+	−	−	−	−
	CVI-623	+		−	−	−	
	CVI-732	+	−	−	−	−	−
**Lost seroprotection post-vaccination**	CVI-256	+	+	−	+/−	+/−	+
	CVI-313	+	+	+	+	+	+
	MHC-031	+		+	+	+	
	P-055	+	+	+	+	+	+
	P-068	+	+	+	+	+	+
	P-134	+	+	+	+	+	+
	CTR-212	+	+	−	−	−	−
	CVI-237	+	+	+	+/−	+/−	−
	CVI-519	+	−	+	+/−	+/−	−
	STM-064	+		+	−	−	
**Lost seroprotection post-infection**	P-164	+	+	+	−	−	−
	VTH-002	+		+	−	−	
	P-014	+		−	−	−	

Differentiated PBMC supernatants from the post-waned timepoints were tested for total IgG antibody levels, as well as IgG antibodies specific to A/H1N1/California/2009 pandemic influenza strain. In addition, the total number of IgG ASCs and the spike-specific ASCs were enumerated from differentiated PBMCs. The + symbol represents the presence of de novo recalled antibodies or ASCs; − represents the absence of de novo recalled antibodies or ASCs, +/− represents that the results were dissimilar based on the two times the experiment was repeated using different PBMC vials. Normal responders seroconverted as expected, nonresponders failed to seroconvert after vaccination, lost seroprotection groups initially seroconverted after vaccination or infection but later lost seroprotective status as shown by a lack of both RBD-binding and SARS-CoV-2 neutralizing antibodies.

## Data Availability

All data is included in the manuscript and its Appendix A, or can be obtained by contacting the corresponding author.

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
