# Peer review of "The Effect of Waning on Antibody Levels and Memory B Cell Recall following SARS-CoV-2 Infection or Vaccination"

_vaccines, 2022, doi:10.3390/vaccines10050696_

Round 1
Reviewer 1 Report
This is an interesting paper in that is confirming that IgG anti RBD neutralising assay titres diminish with time after either infection, vaccination and of combination orders of vaccination and infection. The paper focuses on Humeral and not T-cell response to SARS-CoV2; BUT specifically on neutralisation and anti-RBD assay assessment. There are significant number of samples/participants and this is a great strength of this study. An added feature is that of looking at memory B-cells and which are re-activated.
My major question is what subclasses of antibodies are reactivated from the memory cells. Are they detected in the assay system used and could a bias to memory cells of certain Ig (G) subclasses being highly maintained and not others (i.e. poor memory cell retention of anti-RBD IgG1) explain the dramatic loss, particularly in certain individuals?
Class switching is cytokine driven and maintenance of B-memory cells longterm is often attributed to antigen re-stimulation from germinal centre T-helper cells. Given the functional homologies, presentation of SARS-CoV2 RBD peptides by memory T-helpers and dendritic cells may undergo loss more readily (due to close homology with self - Angiotensin receptors). Whilst other more unique antigen epitopes, but non-neutralising, of the viruses Spike are retained preferentially.
The issue that is lacking is a consideration of the IgG subtype switching that occurs in viral infection. A major consideration is switching to IgG3 (which is highly complement fixing rather than binding neutalization/ ( or effector T cell recruiting). Studies have implicated IgG3 dominance over IgG1 in post infection sera of those with SARS-CoV2 has an association with COVID-19 syndrome (ARDS) development. Thus IgG3 which is complement fixing and may not target the RBD epitope, would not be detected in the functional RBD binding and neutralisation assays. IgG1 anti RBD antibodies would be preferentially detected in these assays. Thus the authors may wish to explore other aspects of the immune response Web in their discussion. And perhaps speculate whether a lack of bias or indeed balance towards IgG1 neutralising antibodies in maturation of memory B cells post vaccination is part of the molecular principles underlying the phenotypes being observed in those with failed "measured" antibody responses?
[HTML] nih.govGlobal characterization of B cell receptor repertoire in COVID-19 patients by single-cell V (D) J sequencing
X Jin, W Zhou, M Luo, P Wang, Z Xu, K Ma… - Briefings in …, 2021 - academic.oup.com … the immune mechanism in COVID-19 patients, we still lack a … the BCR repertoire acrossconvalescent COVID-19 patients. We … Next, we focus on the IgG3 isotype due to its frequent CSR … Save Cite Cited by 5 Related articles All 9 versions [HTML] sciencedirect.com
[HTML] Serological analysis reveals an imbalanced IgG subclass composition associated with COVID-19 disease severity
JL Yates, DJ Ehrbar, DT Hunt, RC Girardin… - Cell Reports …, 2021 - Elsevier … We observed significant increases in both IgG1 and IgG3 with increasing COVID-19 severity,specific for all antigens tested (N, RBD, S1, and S2). The largest difference was seen with … Save Cite Cited by 26 Related articles All 13 versions [HTML] nature.comFull View
[HTML] Immunoglobulin signature predicts risk of post-acute COVID-19 syndrome
C Cervia, Y Zurbuchen, P Taeschler, T Ballouz… - Nature …, 2022 - nature.com … IgG1 was indifferent, whereas IgG3 was higher in both mild and severe COVID-19 … IgG3,whereas severe COVID-19 patients developing PACS failed to show such increase in IgG3 (Fig. … Save Cite Cited by 7 Related articles All 4 versions [PDF] medrxiv.org
Direct detection of humoral marker corelates of COVID-19, glycated HSA and hyperglycosylated IgG3, by MALDI-ToF mass spectrometry
RK Iles, JK IIes, R Zmuidinaite, A Gardiner, J Lacey… - medRxiv, 2021 - medrxiv.org … of COVID-19, contained more glycated HSA and higher mass (glycosylated/glycated) IgG3… respiratory distress syndrome (ARDS) associated with COVID-19. Patients recovering from … Save Cite Related articles All 3 versions [HTML] nature.com
[HTML] Proinflammatory IgG Fc structures in patients with severe COVID-19
S Chakraborty, J Gonzalez, K Edwards… - Nature …, 2021 - nature.com … in IgG3 produced by patients with COVID-19 who were in … COVID-19 was significantlyreduced in core fucosylation when compared with anti-RBD IgG1 from patients with mild COVID-19 … Save Cite Cited by 126 Related articles All 14 versions [HTML] plos.org
[HTML] Immune response dynamics in COVID-19 patients to SARS-CoV-2 and other human coronaviruses
R Ravindran, C McReynolds, J Yang, BD Hammock… - PloS one, 2021 - journals.plos.org
Author Response
We agree with the Reviewer that determining what IgG subclasses are maintained in normal responders, and how their proportions compare to subclasses demonstrated by poor responders would be an interesting research question for future studies. In our ELISA assays, we only detected total IgG antibody binding without distinguishing IgG subclasses. We have included a paragraph in the Discussion section (lines 521-533 in the document with tracked changes) to expand on the limitation of our study pointed out by the Reviewer.
Reviewer 2 Report
This study is informative and should be published. My only concern is regarding how does this study relate to current and future variants of SARS-CoV-2. This could be addressed by adding elements to the Discussion. I don’t think any additional analyses are necessary but discussing how omicron and delta relate to this work would be helpful. For instance, do the authors think the trends observed here would be the same for virus variants? Would waning against the RBD from omicron be shorter than 14 months?
Minor:
Is the SARS-CoV-2 RBD protein used for the ELISA based off USA-WA1/2020 SARS-CoV-2 strain?
Author Response
We thank the Reviewer for raising the intriguing question of variants. We have included text in the Discussion’s limitation section (lines 536-541 in the document with tracked changes) on the topic.
As for the RBD protein, the Reviewer is correct, and we have added the specific strain used for the recombinant RBD protein in the Materials & Methods (line 107).
Reviewer 3 Report
The manuscript by Forgacs and colleagues reports on the evolution of immune response to SARS-CoV2 in infected, vaccinated, or both, over time in a cohort of more than 1,000 individuals from the USA. The authors used ELISA, neutralization and ex vivo methods to evaluate the intensity and the quality of these immune responses.
The duration of antibody response, especially the neutralizing ones, is of paramount interest in the context of repeated vaccinations and of sustained pandemic of COVID-19. The authors are thus commended for that.
The manuscript is well written and the data clearly presented. I have two main queries:
- Given the dataset and the follow up, one would expect an extrapolation, by modelling, of the end point of antibody detectability. That is, the time at which no detectable immune response to SARS-CoV2 RBD is expected. Such a finding would be very useful in a Public Health perspective and could be easily performed.
- The authors are making an equivalence between IgG binding to RDB and neutralization (e.g. page 7, line 262 and onward). It is written “Out of the 25 candidates, 12 either had non-detectable neutralization at the pre-waned timepoint (false positive by ELISA) or still had some level of neutralization potential on the post-waned timepoint (false negative by ELISA) (Figure 4B).” These are two different things and the authors modify this assumption to make it clear that biding and neutralization do not map systematically.
Author Response
- We really appreciate the Reviewer bringing up the valuable point about extrapolating antibody loss. Based on extrapolating the antibody loss based on the decay curves, we see the following pattern:
Modeling suggests that naïve, vaccinated participants will sero-revert to antibody negative as early as 17 months after vaccination. Infected, vaccinated participants will not have significantly higher concentration of antibodies than infected, unvaccinated participants 17 months after vaccination, and sero-revert by 22 months after vaccination. However, these numbers are almost certainly severe underestimates as we already have data from participants up to 15 months past full vaccination that shows the decay rate become less and less steep each month (a phenomenon we allude to in the manuscript in lines 402-405 and 417-420 in the document with tracked changes), and there is no sign of full-scale sero-reverting over the course of the next 2 months’ time. Thus, we believe reporting those numbers would be alarmist and misleading, and we hope the Reviewer agrees with that sentiment. However, we are glad that the Reviewer picked up on these trends and we could share them with the Reviewer. - As stated in the sentence prior to the one the reviewer mentioned (now lines 312-314), “In order to confirm that these participants have in fact lost seroprotective status, viral neutralization tests using infectious SARS-CoV-2 USA-WA1/2020 strain were performed.” However, we acknowledge that in later parts of the manuscript we do not stress enough that these participants have lost both binding (confirmed by ELISA) and neutralizing (confirmed by VN) antibodies. We clarified in lines 464-466 in the Discussion.
Reviewer 4 Report
The paper analyzes anti-RBD antibody levels after vaccination or after infection in an ample cohort of subjects studied over 12-14 months after infection or vaccination (including third dose) with mRNA vaccines. In a small group that lost protective antibodies memory B cells were also analyzed.
The paper contains many interesting data, well presented and adequately discussed.
Only minor points can be raised.
- The choice of R848 for B mem expansion should be briefly explained.
- Antibody specificity in “false negative” sera (that contain neutralizing antibodies but no anti-RBD antibodies) should be discussed.
Author Response
- An explanation for the use of R848 and rIL-2 has been added to the Methods section (lines 135-138 in the document with tracked changes)
- We appreciate the Reviewer’s request for the discussion of false negative ELISA results, some possible explanations are now included in the Discussion section lines 466-473.
Reviewer 5 Report
The manuscript “The effect of waning on antibody levels and memory B cell re-call following SARS-CoV-2 infection or vaccination” by Forgacs and colleagues is a primarily descriptive clinical study comparing RBD specific antibody levels after vaccination compared to natural infection over 14 months. Nevertheless, this is an important contribution to our understanding of the long term antibody mediated immune response after natural infection and after vaccination. Overall, the manuscript is clear.
However, there are a few issues that can be easily addressed.
Specific critique
Abstract
Line 22 rephrase 70% of the vaccinated when assessing 13 patients. Instead specify how many out of 13 had the significant recall of memory B cells.
Introduction:
Lines 36/37: T cell responses are also important. Include a statement to this fact so the reader is not misguided. Later on, the authors mention cellular immunity but still only mention B cells and plasma cells which are normally considered part of the humoral response.
Line 45: An individual, not a individual.
Lines 75/76: this should be reworded- antibodies do not trigger recall. It is suggested that the authors rather talk about whether presence of specific antibody titers is indicative of a strong recall response.
Materials and methods:
The uneven distribution of race and ethnicity imposes limitations of the study. Furthermore, it does not reflect the current distribution among the population Los Angeles and other areas.
Results:
Lines 268/269: Serum does not contain cells and cannot differentiate. Please refer to peripheral blood derived cells.
S1 and Table S3 should be included in the main text along with the information regarding immunosuppressive conditions.
Table 1:
The green and red box are confusing (red is such a bold color and does not reflect “no reaction”. I think + and – signs might be clearer.
Figures:
For each figure, clearly state in the legend which out of the 4 patient groups (naïve unvaccinated, infected unvaccinated, naïve vaccinated, infected vaccinated) is depicted. E.g. the legend in Figure 3 only refers to 3rd dose and does not clarify whether prior to vaccine a vaccination had been given.
Figure 4:
Perhaps use open symbols to better detect the individual responses or show each symbol for the 13 subjects- only 7 symbols can be differentiated in Figure 4B.
Author Response
We thank the Reviewer for their thoroughness. Here we address the issues raised point-by-point:
Abstract: We have changed the Abstract to reflect actual numbers instead of percentages (line 23 in the document with tracked changes).
Introduction: A sentence on the importance has been added to lines 58-60 of the Introduction and the term humoral immunity was avoided in line 57. We appreciate the Reviewer catching the typo and suggesting rewording which we implemented in lines 48 and 81, respectively.
Materials and methods: We have included a sentence on how race/ethnicity imposed limitations on our findings in the limitations section of the Discussion (lines 549-551)
Results: Naturally the Reviewer is right about serum not containing cells, we have corrected the sentence to portray our intended meaning (line 320)
While we agree that Figure S1 and Table S3 are important, we do not believe that they are necessary for the understanding of the manuscript. As the manuscript already has 5 figures and 1 table, we would prefer to keep those in as Supplemental, unless the Reviewer insists upon their inclusion and the Editor allows for it.
Table 1: The style of Table 1 has been changed based on the Reviewer’s recommendations.
Figures: For Figure 3 depicting the reactions to the 3rd vaccines dose, participants were not separated based on pre-immunity due to the low number of pre-immune boosted participants at the time. A statement to that effect was included in the Figure 3 caption (lines 300-301).
Figure 4: The Reviewer is correct in pointing out that there are several overlapping participants with the same endpoint titers represented in Figure 4B. Due to the high number of overlapping data points in some cases, we opted to include the number of overlapping data points next to the dots and included an explanation to that effect in the caption (lines 345-346).